# Reformulation of Packaged Foods and Beverages in the Colombian Food Supply

**DOI:** 10.3390/nu12113260

**Published:** 2020-10-24

**Authors:** Caitlin M. Lowery, Mercedes Mora-Plazas, Luis Fernando Gómez, Barry Popkin, Lindsey Smith Taillie

**Affiliations:** 1Department of Nutrition, Gillings School of Public Health, University of North Carolina at Chapel Hill, Chapel Hill, NC 27599-7400, USA; clowery@unc.edu (C.M.L.); popkin@unc.edu (B.P.); 2Departamento de Nutrición Humana, Universidad Nacional de Colombia, Bogotá, Carrera 45 N°26-85, Bogotá 11001, Colombia; mmorap@unal.edu.co; 3Facultad de Medicina, Pontificia Universidad Javeriana, Bogotá, 8 piso Hospital Universitario San Ignacio, Bogotá 110231, Colombia; l.gomezg@javeriana.edu.co; 4Carolina Population Center, University of North Carolina at Chapel Hill, Chapel Hill, NC 27516-2524, USA

**Keywords:** Latin America, obesity prevention, food industry, food labeling, food policy

## Abstract

Public discussion, advocacy, and legislative consideration of policies aimed at reducing consumption of processed foods, such as sugar-sweetened beverage (SSB) taxes and mandatory front-of-package (FOP) warning labels, may stimulate product reformulation as a strategy to prevent regulation. In Colombia, there have been major legislative pushes for SSB taxes and FOP labels, although neither has passed to date. In light of the ongoing policy debate and successful implementation of similar policies in Peru and Chile, we explored manufacturer reformulation in the Colombian food supply. We compared the quantities of nutrients of concern (including sugar, sodium, and saturated fat) from the nutrition facts panels of the same 102 packaged foods and 36 beverages from the top-selling brands in Colombia between 2016 and 2018. Our analyses showed a substantial decrease in median sugar content of beverages, from 9.2 g per 100 mL to 5.2 g per 100 mL, and an increase in the percentage of beverages containing non-nutritive sweeteners (NNS), from 33% to 64% (*p* = 0.003). No meaningful changes in the quantities of nutrients of concern among foods were observed. Our findings suggest little reformulation has occurred in Colombia in the absence of mandatory policies, except for the substitution of sugar with NNS among beverages.

## 1. Introduction

In the past few decades, Colombia has undergone a nutrition transition, marked by rising levels of obesity [1,2] and increased availability and consumption of sugar-sweetened beverages (SSBs) and other ultra-processed foods [3,4], which contain ingredients generated through industrial processes and are characterized by hyper-palatability, convenience, and profitability [5,6]. This shift is part of a larger global trend, linked to rising rates of type 2 diabetes and cardiovascular disease [7,8]. Several Latin American countries have instituted policies aimed at reducing consumption of energy-dense, nutrient-poor foods [9], such as sugar-sweetened beverage (SSB) taxes and mandatory front-of-package (FOP) warning labels [10,11,12,13,14,15,16], which inform consumers if a product is high in specific nutrients of concern (e.g., sugar, saturated fat, sodium).

While evaluations of SSB taxes and FOP labels have primarily focused on customer perceptions, purchases, and consumption, these policies have the potential to influence diet and health through another pathway: manufacturer reformulation [16,17,18,19]. Threshold-based SSB taxes and FOP labels provide manufacturers with an incentive to reformulate their products to avoid paying a tax or being subject to a mandatory warning label. From a public health perspective, reformulation may be a desirable “upstream” intervention to improve diet quality without requiring individuals to change their eating behavior [20,21], although many reformulated products may not represent a major improvement in food quality [16,22,23]. Evidence from impact studies of the U.K.’s Soft Drinks Industry Levy (SDIL) [24,25] and Chile’s mandatory Law of Food Labeling and Advertising [26] suggests that these policies have led to substantial manufacturer reformulation of packaged products. The purpose of this paper is to explore manufacturer reformulation in Colombia, a country that is actively considering policies similar to those of nearby Peru and Chile, and that shares many of the same food suppliers with those nations.

In Colombia, there have been major legislative and public media pushes for FOP warning labels, SSB taxes, and restrictions on unhealthy food marketing to children. In 2016, Colombia attempted to pass “healthy taxes” on tobacco and SSBs as part of a tax reform bill, but industry opposition led to the elimination of the SSB tax from the final proposal [27]. In the midst of the SSB tax debate, the beverage industry signed a “responsible self-regulation” agreement, promising to limit advertising and sales of unhealthy beverages in schools and to provide smaller portion sizes and low-calorie beverage options, among other commitments [28]. In January 2019, Colombia subjected soft drinks to a multi-phase value-added tax (VAT), nullifying an earlier provision that had taxed beverages at only one stage in the supply chain. The expected price increase from this change is small, and the impact has not yet been evaluated. In terms of food labeling, the majority of food products available in Colombian stores have a nutrition facts label, but they are only legally required if the packaging contains a nutrition or health claim [29]. Recently, the Ministry of Health unveiled a proposal for mandatory circular FOP warning labels, requiring labels on products high in added sugar, saturated fat and sodium [30,31].

Although Colombia has not yet passed a specific SSB tax or mandatory FOP labeling requirement, one important question is whether the ongoing legislative debate, mass media campaigns highlighting the harms of sugary drinks, and the beverage industry’s voluntary commitment might drive manufacturer reformulation. One possibility is that manufacturers would be incentivized to reformulate as a strategy to avoid or argue against governmental regulation [23]. Additionally, we were interested in potential spillover effects from existing policies in Chile and Peru that may have led to manufacturer reformulation, including SSB taxes and mandatory FOP warning labels. Many multinational companies offer products throughout Latin America, so if they reformulated their products for Chile and Peru, it is possible that they may offer these reformulated products in Colombia. Few studies have examined the impact of national policies on the food supply in neighboring countries, but limited evidence suggests that trans fatty acid (TFA) policies in Denmark and the U.S. led to reductions in TFAs in Scandinavia and Costa Rica [32,33,34,35]. Global companies have demonstrated that they have the technology to reduce sugar and sodium in regulated contexts like Chile [26], but it is unclear if they sell these reformulated products in countries without regulation. Prior research suggests that packaged foods in middle-income countries like Colombia, which have fewer nutritional regulations, may be of lower nutritional quality than similar products in high-income countries [36,37]. Finally, little is known about current trends in product reformulation in Colombia. A systematic analysis of existing product reformulation may inform the development of future policies aimed at reducing nutrients of concern in the processed food supply.

In this paper, we analyze trends in product reformulation among packaged food and beverages from the top-selling brands in Colombia between 2016 and 2018. We explore changes in the quantities of nutrients of concern, which were selected based on two nutrient profile models frequently used to develop and inform food policies in the Latin American region. We used the nutrient profile models to identify products that would be subject to regulation under a warning label law, as manufacturers would have the most incentive to reformulate these products to avoid the regulation, which is particularly salient at present, in light of the Ministry of Health’s FOP proposal [31,38].

## 2. Materials and Methods

The Nutrition Facts Panel (NFP) data of more than 15,000 packaged foods and beverages were collected from 16 stores belonging to the five largest supermarket chains in Bogota, Colombia, in 2016 and 2018 using the Kanter photographic method [39], as described in prior studies in Chile and Colombia [40,41,42]. The stores were located in neighborhoods of low, medium, and high socioeconomic status (SES) across the city, which has a population of nearly 11 million people [43]. NFP and package data (e.g., barcodes) were entered into a REDCap (Research Electronic Data Capture) database hosted at the University of North Carolina at Chapel Hill [44]. Nutrient quantities were standardized to the quantity per 100 g or 100 milliliters of the product by trained nutritionists.

### 2.1. Nutrient Profile Systems

Because we are interested in the impact of proposed policy interventions on the food supply, we selected nutrients for evaluation based on two nutrient profiling models. The Pan American Health Organization (PAHO) and the Chilean nutrient profile systems were chosen because they have been discussed as the basis for policy regulations in Latin America, including in Colombia [11,12,40,45]. We used these models to identify products that would be regulated if an FOP warning label policy were passed in Colombia. Both models use nutrient density thresholds for regulation, defined as the amount of a given nutrient per 100 g or 100 mL of the product.

Under the PAHO model, foods classified as unprocessed or minimally processed were exempt. Products were considered processed/ultra-processed if the ingredient list included any added sugar, sodium, saturated fat, and/or non-nutritive sweetener (NNS) [45]. Processed and ultra-processed foods were considered regulated if (1) free sugars contributed ≥10% of total energy; (2) total fat contributed ≥30% of total energy; (3) saturated fat contributed ≥10% of total energy; (4) trans fat contributed ≥1% of total energy; (5) sodium content was ≥1 mg/kcal; or (6) the product contained any NNS [45]. Products containing NNS were identified based on the ingredient list, since manufacturers are not required to disclose the quantity of NNS in their products. If free sugar was not reported on the package, it was estimated from total sugar using a free sugar factor (range: 0 to 1) by registered dietitians, using the algorithm created by the Expert Consultation Group of PAHO Nutrient Profile Model [45]. Products could receive up to 6 labels under this model, warning consumers about the total fat, saturated fat, trans fat, free sugar, sodium, and/or NNS content of the product.

The Chilean Law of Food Labeling and Advertising used a phased approach to establish nutrient guidelines, with stricter limits over time. For the present study, regulation status under the Chilean model was based on the third and final set of nutrient density standards, implemented in 2019 [11]. The Chilean nutrient profile model exempts foods without added nutrients of concern. The model applies different standards for liquids, defined as products reporting servings in mL, and solid food, operationalized as products measured in grams. Solid food products were considered regulated if they contained (1) >10 g of total sugar and included an added sugar ingredient; (2) >4 g of saturated fat and included an added saturated fat ingredient, (3) >400 mg sodium and included an added sodium ingredient, or (4) >275 kcal per 100 g of the product and included an added sugar, saturated fat or sodium ingredient. Liquids were subject to the same standards with regard to added nutrients of concern, but the regulation thresholds were lowered to (1) >5 g of total sugar, (2) >3 g of saturated fat, (3) >100 mg of sodium, or (4) >70 kcal per 100 mL of the product. Products could receive up to 4 labels under this model, warning consumers that the product is high in calories, saturated fat, total sugar, and/or sodium.

### 2.2. Sample Selection

#### 2.2.1. Beverage Selection

In order to select a sub-sample of the items for the present study, we used Colombian market share data from Euromonitor International [46], a global market research company, to identify the top-selling brands in 16 beverage categories (up to 3 brands per category), for a total of 41 category-specific brands (29 unique brands) (see Appendix A, Figure A1). The Euromonitor market share data provided the annual percentage of sales belonging to a given brand in each category. The selected brands represented more than 50% of the market in each category (mean: 82.3–84.9%), except for milk substitutes, as Euromonitor only had data on one brand in both years (representing 24.7–24.9% of the market) (see Appendix A, Table A1). Because of the need to identify individual products to compare at each time point, private label (0.0–2.7% of the market share of each category) and unidentified, “other” brands were excluded. Unspecified “other” brands generally made up a small percentage of market sales and thus would not have been included in our sample even if they represented a single, identifiable brand, except in the cases of ready-to-drink (RTD) Coffee, RTD Tea, Flavored Milk, and Milk Substitutes.

Because the categories were highly specific, we selected one item per brand within the given category to obtain a sub-sample of items for the present study (e.g., within the category “Ginger Ale,” the brand “Schweppes” was a top-seller, from which we selected the product “Schweppes Ginger Ale (1.5 L)”). For analysis, the 16 beverage categories were collapsed into seven groups: Carbonates (Regular), Concentrates, Diet Colas, Energy/Sports Drinks, Ready-to-Drink (RTD) Coffee/Tea, Flavored Milk/Milk Substitutes, and Juice Drinks/Nectars. We did not include plain milk, 100% juice, or (unsweetened) bottled water in our 16 categories because they are exempt from most FOP labels and SSB tax policies and thus unlikely to be reformulated.

Because this study focuses on product reformulation, we wanted to identify the same products for comparison in both years. If the same product, based on product ID (UPC) and product name, was available in the same size in 2016 and 2018, they were considered a matched pair. The final sample contained 36 beverages, as five of the 41 category-specific brands did not have any beverages in the relevant category in the NFP dataset in both years and were excluded. If multiple matched pairs were available in both years, we randomly selected one. If the product was not available in the exact same size both years, we paired the 2016 item with the nearest available size of the same product in 2018 (*n* = 8). However, nutrient content was standardized to the quantity per 100 mL for all beverages, so differences in package size are not expected to impact nutrient composition. Among products with multiple flavors, such as juice drinks and sports drinks, we selected matched pairs from the flavor that was available in the largest number of sizes as a proxy for popularity. If there were an equal number of size options for each flavor, we randomly selected one flavor, as we do not expect reformulation status to vary by flavor within the same product type.

#### 2.2.2. Food Selection

We followed the same process to identify foods, although we selected up to five top-selling brands per food category (rather than three), provided each brand accounted for at least 5% of sales within the specified food category in both years. This provided a total of 53 brands across 14 food categories (see flowchart in Appendix A, Figure A1). Because the market for food is less concentrated than for beverages, the total market share of each category that was represented by the selected brands varied from 16.1–17.3% for Baked Goods to 74–74.6% for Soups (details in Appendix A, Table A2). Mean market share per category ranged from 44.6% (2016) to 44.7% (2018). Two of the 16 Euromonitor packaged food categories were excluded prior to brand identification because they were comprised of products exempt from the PAHO and Chilean regulations (edible oils and baby food). One category (Dairy) was restricted to two sub-categories (Cheese and Yogurt), as the other sub-categories primarily comprised culinary ingredients and thus were exempt from most regulations (Butter, Other Dairy) or included in Beverages (Drinking Milk). We then identified food products from top-selling brands from each category in the NFP dataset. Example products can be found in Appendix A, Table A3. Product identification was restricted to packaged foods containing any added nutrient of concern (salt, sugar, or saturated fat) or non-nutritive sweetener (NNS) as single-ingredient foods such as culinary ingredients or unprocessed foods are exempt from most regulations. Seven of the 53 category-specific brands did not have any eligible items in the relevant category in the NFP dataset in both years and were eliminated, which resulted in the exclusion of three food categories because they contained fewer than 5 matching, non-exempt products (Processed Fruits and Vegetables, Sweet Spreads, and Confectionary). The final food sample was selected from 46 (category-specific) brands in 11 categories.

We selected up to ten products in each category from the top-selling brands because of the greater diversity of product offerings within each category as compared to beverages. After dropping one product because the label did not report the quantities of multiple nutrients of concern, our final sample contained 102 food products. When possible, we selected the same number of products from each brand in a category (e.g., 5 products per brand); in the event of uneven division, the remainder went to the brand(s) with the larger market share. As with beverages, products were matched on product ID, product name, and size, when possible. For items with multiple varieties, we chose the matched pair available in the largest number of sizes, or if there were an equal number of size options, we randomly selected one product.

Prior to evaluating regulation status, all food and beverages in powder form (soups, beverage concentrates) were reconstituted based on the package instructions, and nutrient densities were calculated per 100 mL of the reconstituted product.

### 2.3. Statistical Analyses

We compared quantities of nutrients of concern per 100 g/mL, including total fat, saturated fat, trans fat, sodium, total sugar, free sugar, and energy density, overall and by food group, in the same 138 items in 2016 and 2018. We also analyzed the proportion of each sample that contained NNS, as well as the proportion that would receive at least one label under the PAHO guidelines and the Chilean regulation criteria. Finally, we compared the mean number of labels that products would receive under each model in 2016 and in 2018.

We used the Wilcoxon signed-rank test, a non-parametric test for paired data, to test for differences in nutrient density by year, as the data were not normally distributed. We present the median and interquartile range (IQR) for the nutrient density variables. To evaluate changes in the proportion of products containing NNS and the proportions of products regulated under the two NFP models, we used McNemar’s test for paired data [47]. Analyses were performed in Stata 16 [48]. Alpha was set at 0.05.

## 3. Results

Table 1 presents the median quantities of the nutrients of concern regulated under the Chilean policy by year. Between 2016 and 2018, median calories in beverages declined from 41.7 (IQR: 25–45) to 25.0 (IQR: 4.6–42.7) kcal (*p* < 0.001), while median calories in food products remained relatively stable. Sodium in beverages increased very slightly, although the sodium content of beverages was low. Changes in saturated fat content were not statistically significant for foods or beverages. Results by food group can be found in Appendix A, Table A4. Changes in median quantities of nutrients of concern were not significant for any food group. Appendix A, Table A5 shows changes in median quantity of nutrients of concern based on the PAHO model, including trans fat and total fat. There were no changes in the number of products containing trans fat (*n* = 5) or the amount of trans fat they contained. Median total fat remained the same among beverages alone, but declined slightly among food products, from 7.5 (IQR: 1.0–18.5) to 7.0 (IQR: 0.8–17.3) g/100 g (*p* = 0.037).

Median total sugar content among beverages decreased from 9.2 (IQR: 5.8–10.4) to 5.2 (IQR: 1.0–8.7) g/100 mL (*p* = 0.002) (see Table 2). Median sugar content in foods remained approximately the same. Free sugar content in beverages declined from 7.8 (IQR: 4.5–10.2) to 5.0 (IQR: 1.0–7.5) g/100 mL (*p* = 0.003) (Table 2). The decreases in sugar content were accompanied by a substantial rise in the proportion of beverages containing non-nutritive sweeteners, from 0.33 to 0.64 (*p* = 0.001). The proportion of foods containing NNS was largely unchanged.

Under the PAHO model, 99.3% of the sample (100% of beverages and 99% of foods) would be subject to at least one warning label, a proportion that did not change between 2016 and 2018 (Figure 1a). The mean number of warning labels increased from 2.17 to 2.28 (*p* = 0.010), primarily due to the increase in the number of beverages containing NNS. NNS is not regulated under the Chilean model. Under the Chilean model, the percentage of beverages that would receive at least one label decreased from 78% to 50% (*p* = 0.002), while the percentage of foods that would receive a label remained approximately the same (92% to 90%) (Figure 1b). The mean number of labels a product would receive under the Chilean regulations also decreased, from 1.57 to 1.46 (*p* = 0.006). The mean number of labels to which beverages would be subject declined from 0.86 to 0.58 (*p* = 0.002). There was no significant change in the mean number of labels to which food products would be subject.

## 4. Discussion

While we know from Chile and other high-income countries that food manufacturers are able to reformulate their food products to remove nutrients of concern, we did not find evidence that these reformulated products are being marketed in Colombia. While calorie density and sugar content declined in beverages, there was minimal change among foods, and little difference in the quantity of other nutrients of concern such as sodium, trans fat, and saturated fat, among foods or beverages. The lack of product reformulation suggests that the ongoing legislative debate over FOP warning labels, marketing restrictions, and SSB taxes in Colombia, as well as the implementation of similar mandatory nutrition policies in nearby countries like Chile and Peru, have not had significant effects on the food supply in Colombia. This is perhaps unsurprising given existing evidence that in low and middle-income countries (LMICs), which generally have fewer nutrition regulations and less enforcement, packaged foods contain higher quantities of nutrients of concern than similar products in high-income countries [36,37]. In order to generate substantive changes to the food supply, comprehensive regulations, such as mandatory FOP warning labels or bans on potentially harmful compounds like artificial trans fatty acids, may be required.

The only meaningful change we observed between 2016 and 2018 was a decrease in the sugar content of beverages, which also led to a reduction in calorie density among drinks. There are several possible drivers of beverage reformulation in Colombia. First, the unsuccessful attempt to pass an SSB tax in 2016 (and subsequent proposals for taxes, FOP labels, and marketing restrictions) may have motivated the industry to reformulate beverages, in anticipation of the potential enactment of such legislation. SSB taxes may lead to anticipatory reformulation, as in the U.K. [24,25], although no studies to date have looked at the impact of unsuccessful beverage tax proposals on product reformulation. While Colombia’s shift from a mono-phase VAT to a multi-phase VAT on soft drinks in 2019 will likely raise beverage prices slightly, the impact is expected to be small.

Second, the beverage industry in Colombia agreed to limit advertising and sales of unhealthy beverages in schools and to provide smaller portions and diet beverage options in 2016 [28]. Industry self-regulatory commitments may spur product reformulation as a means to appear healthier than competitors [49,50], signal commitment to public health [51], and avoid mandatory governmental regulation [23,52,53,54]. A 2017 report from think tank Dejusticia found that schools in Bogota were unaware of the pledges, and most sold beverages not meeting the pledge standards [55]. Nevertheless, it is possible the changes occurred as a result of the commitment.

Third, the reformulation seen in Colombia could be a spillover effect from policies in Chile and Peru, which have both increased taxes on SSBs and implemented FOP warning labels in recent years. The absence of any effect on food and the lack of meaningful change in other nutrients of concern calls into question the FOP labels as a primary driver. Nonetheless, many beverage companies are multinational corporations and could choose to sell products they reformulated for the Chilean context elsewhere in the region. Finally, reformulation could be driven by changing consumer demands at the local, regional, or global level. The push for an SSB tax in 2016 was linked with significant mass and social media campaigning about the negative health effects of sugary drink consumption, as were subsequent marketing restriction and warning label proposals.

Our findings have significant implications for a current FOP policy under consideration by the Colombian Ministry of Health. FOP labels could have a large impact on the Colombian food supply, as the labels would apply to 66.4–80.2% of packaged food products on the market [40]. Because we restricted our sample to packaged products containing added nutrients of concern (sugar, salt, saturated fat) or NNS, the proportion of products in our study that would receive labels was higher, although we also found that the PAHO nutrient profile model resulted in a greater proportion of regulated products than the Chilean model [40]. The choice of nutrient profile model may impact the amount and type of reformulation that occurs. Based on the PAHO model, we found no change over time in the percentage of products that would be regulated, which in our sample was nearly 100%. We observe a slight increase in the number of labels each product would receive over time, due to the increase in the number of beverages receiving an added NNS label. Because the PAHO model regulates NNS, manufacturers would likely have less incentive to reformulate products to reduce sugar. Under the Chilean nutrient profile model, fewer products would have received labels at both time points, with a significant reduction between 2016 and 2018. The Chilean model, which does not regulate non-nutritive sweeteners, would have allowed half of all beverages in our sample to avoid a warning label in 2018.

Without mandatory regulation, we see minimal change in the quantities of nutrients of concern among food products in Colombia. Findings from Chile suggest that mandatory food labeling policies could have larger effects on the food supply among both food and beverages, as a recent evaluation of the Law of Food Labeling and Advertising found reductions in sugar not only among beverages and milk, but also among breakfast cereals, sweet baked goods, and sweet and savory spreads [26]. The study also identified reductions in the salt content of high-sodium food categories such as cheeses, soups, and sausages [26], which could potentially reduce hypertension and cardiovascular disease. While this study found reformulation among a sub-sample of beverages from the top-selling brands in Colombia, these products may not be widely available in all regions of the country and do not reflect changes in brands not included in this analysis. An SSB tax may spur further reformulation. In the U.K., manufacturer reformulation began after the announcement of the SDIL, with the percentage of beverages above the lower tax threshold (5 g/100 mL) declining by 19.5%. However, after the implementation of the levy, the percentage of products above the lower tier fell further (−30.7%) [25]. Additionally, these policies have multiple paths of impact, not limited to manufacturer reformulation, including financial disincentives and highlighting health harms, which may influence customer purchases and consumption and thereby amplify population health benefits.

This study has several strengths. First, it adds to the literature on product reformulation in the food supply. Measuring changes in the food supply presents challenges as databases such as food composition tables may not be updated frequently enough to evaluate changes and may not be sufficiently comprehensive [56]. This study used data methodically collected by photographing the nutrition labels of products in Colombian supermarkets at two timepoints, according to the same procedure developed in Chile. While our sample of products was small, they were selected from the top-selling brands in each category by market share. Products were matched on name and barcode (UPC) to ensure comparison of the same product in each year.

We acknowledge several limitations. First, we included a limited number of products. Products were selected based on market share, but several food categories encompassed a broad array of products and those selected may not be entirely representative of the entire category. Second, while brands were identified based on national sales data from Euromonitor, specific products within each brand were selected based on their availability in a dataset of Nutrition Facts Panel data collected from supermarkets in Bogota. While most products in our sample came from multinational or national brands, they may not be representative of the products found outside of the capital. Third, products had to be available in both 2016 and 2018 and were selected from matched pairs primarily identified via product name and barcode. If manufacturers assigned new barcodes to some reformulated products, this may have led to underestimation of the true extent of reformulation. Finally, reformulation is treated as a positive outcome to improve diet quality without necessitating behavior change at the individual level [16,20,21]. However, many reformulated products are ultra-processed and may contain other potentially unhealthy ingredients not removed in the reformulation process [22]. Further study is needed to examine the health impacts of product reformulation, particularly in light of the widespread replacement of sugar with NNS among beverages.

## 5. Conclusions

This study found little evidence of reformulation to reduce nutrients of concern among the top-selling foods in Colombia, despite active, sustained political and societal debate around obesity prevention policies. However, sugar content of beverages decreased over time, as manufacturers replaced sugar with non-nutritive sweeteners. This study highlights the need for comprehensive, mandatory policies including FOP warning labels to reduce nutrients of concern in the national food supply.

## Figures and Tables

**Figure 1 nutrients-12-03260-f001:**
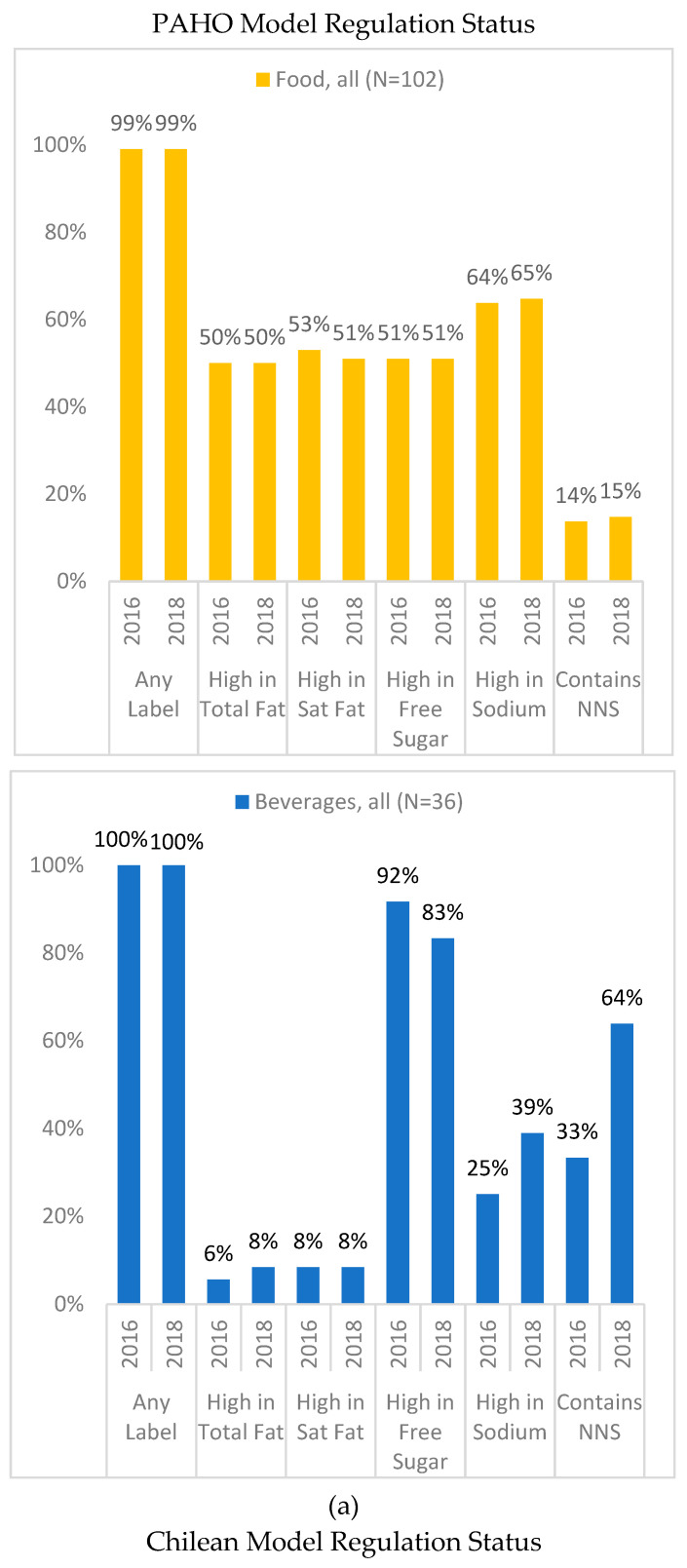
(**a**) Products that would be regulated under the Pan American Health Organization (PAHO) nutrient profile model by year. Trans fat is not included in this figure as there was no change between 2016 and 2018. Non-nutritive sweeteners (NNS) are sugar substitutes which contain few or no calories. (**b**) Products that would be regulated based on the nutrient profile model used in the Chilean Law of Food Labeling and Advertising by year.

**Table 1 nutrients-12-03260-t001:** Median (IQR) quantity per 100 g/mL of selected nutrients of concern in 2016 and 2018 ^1,2^.

Category	Calories (kcal)		Saturated Fat (g)		Total Sugar (g)		Sodium (mg)	
Year	2016	2018	2016	2018	2016	2018	2016	2018
Food, *n* = 102	269.7	273.9	3.3	3.2	6.9	7.6	400.0	381.0
	(244.4)	(241.3)	(9.3)	(8.9)	(20.0)	(20.0)	(536.7)	(533.3)
Beverages, *n* = 36	41.7	**25.0 *****	0.0	0.0	9.2	**5.2 ****	10.2	12.5
	(20.0)	(37.1)	(0.0)	(0.0)	(4.6)	(7.7)	(25.0)	(21.3)

^1^ Parenthetical values represent the interquartile range (IQR). ^2^ Bolded 2018 values differ significantly from 2016 values. We used exact *p*-values due to small sample size. ** *p* ≤ 0.01, *** *p* ≤ 0.001.

**Table 2 nutrients-12-03260-t002:** Median (IQR) quantity of selected nutrients of concern per 100 mL by beverage type ^1,2^.

Category	Calories (kcal)		Total Sugar (g)		Free Sugar (g)		Proportion with any NNS ^3^	
Year	2016	2018	2016	2018	2016	2018	2016	2018
Beverages, all (*n* = 36)	41.7 (20.0)	**25.0 ***** (37.1)	9.2 (4.6)	**5.2 **** (7.7)	7.8 (5.6)	**5.0 **** (6.5)	0.33	**0.64 *****
Carbonates (*n* = 12)	41.7 (1.7)	**29.2 *** (28.3)	10.2 (0.9)	**7.5 **** (6.3)	10.2 (0.9)	**7.5 **** (6.3)	0.08	**0.58 ***
Concentrates (*n* = 3)	4.9 (17.4)	5.0 (17.4)	1.5 (4.0)	1.5 (4.5)	1.5 (4.0)	1.5 (4.5)	1.00	1.00
Diet Colas (*n* = 3)	0.0 (0.0)	0.0 (0.0)	0.0 (0.0)	0.0 (0.0)	0.0 (0.0)	0.0 (0.0)	1.00	1.00
Energy/Sports Drinks (*n* = 5)	25.0 (0.0)	25.0 (4.2)	5.8 (0.4)	5 (1.3)	5.8 (0.4)	5.0 (1.3)	0.40	0.40
Juice Drinks/Nectars (*n* = 6)	43.8 (8.3)	30.8 (31.8)	9.2 (2.0)	6.5 (7.9)	4.9 (4.9)	3.4 (3.0)	0.33	0.67
Flavored Milk and Milk Subs (*n* = 3)	85.0 (50.0)	75.0 (55.0)	9.5 (8.5)	9.0 (9.0)	4.8 (2.5)	4.5 (2.8)	0.33	0.67
Ready-to-Drink Coffee/Tea (*n* = 4)	37.8 (17.3)	27.1 (34.1)	9.3 (1.5)	6.3 (6.0)	9.3 (1.5)	6.3 (6.0)	0.00	0.50

^1^ Parenthetical values represent the interquartile range (IQR). ^2^ Bolded 2018 values differ significantly from 2016 values. We used exact *p*-values due to small sample size. * *p* ≤ 0.05, ** *p* ≤ 0.01, *** *p* ≤ 0.001. ^3^ Manufacturers are not required to declare the amount of non-nutritive sweetener (NNS) in products, so we used McNemar’s test to analyze the proportion of products containing any NNS, based on the ingredients list.

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
