# Peer review of "Reformulation of Packaged Foods and Beverages in the Colombian Food Supply"

_nutrients, 2020, doi:10.3390/nu12113260_

Round 1

Reviewer 1 Report

Abstract:

Lines 17-18. How product reformulation can be strategy to prevent regulation?

Lines 26-27. Authors claim “no meaningful changes (what is that mean?)in the quantities of nutrients (…)”, and in lines above pointed the differences in sugar content (decline in 40%).

Introduction:

Line 33: No clear definition of what is “ultra-processed foods”. I would advise to give a brief definition.

Monteiro, CA, Cannon, G, Moubarac, J-C et al. (2017) The UN Decade of Nutrition, the NOVA food classification and the trouble with ultra-processing. Public Health Nutr: “‘The term “ultra-processed” was coined to refer to industrial formulations manufactured from substances derived from foods or synthesized from other organic sources. They typically contain little or no whole foods, are ready-to-consume or heat up, and are fatty, salty or sugary and depleted in dietary fibre, protein, various micronutrients and other bioactive compounds. Examples include: sweet, fatty or salty packaged snack products, ice cream, sugar-sweetened beverages, chocolates, confectionery, French fries, burgers and hot dogs, and poultry and fish nuggets”

Not clear if Colombia had applied the mandatory labelling requirements for food and beverages incl. law (taxes) (line 57 “in January 2019, Colombia subjected soft –drinks to  a multi-phase value added tax..”) or not (line 63: “not yet passed an SSB tax”).

It would be appreciate if authors would precise what are mandatory information on a label of a product according to Colombian law.

Materials and methods:

Lines 91-92: Why all product’s nutrients composition was “standardized to 100 g/ml by trained nutritionists”? Isn’t a mandatory in Colombia to give such an information on a label (nutrient value in EU must be given in 100 ml/g), if not it should be explained for readers apart from Americas.

Another question refers to this process: does the nutritionist examined the product (chemical, physical analysis) or just counted the value from the label.

Beverage selection:

Line 112: Why did the authors settled the necessity to compare the same packaging volumes for a given product? If it’s the same product (same composition) package volume is not influencing the nutrient value.

Food selection

I would advise to start this paragraph form a sentences placed in line 135-description of choosing process. Then description what did authors selected.

PAHO – lack of full name

I would also advise to set there a table with explanation of what were the products (sort, kind). E E.g. locating a regular bun in a same group with long shelf life cake (packed) can be misleading. Or e.g. Where were located: ready–to-eat salads prepared with added substances (sugar, salt etc.)?

Nutrient profile system:

Authors described the regulation’s demanding and at the end of a paragraph they mentioned 4 labels. It would be more clear it they would present this labels.

Since ‘Nutrients’ is worldwide available journal, authors should explain why in south America is using Sodium (Na+) instead of salt (NaCl) as a definition of a factor that may causes hypertension and heart diseases. Such a definition was used in a past in EU but few years ago it was substituted (again) with salt (NaCl) to be more understandable for consumers.

Results:

Is the table 1 title the results?(lines 201-214)

Why did authors gave the median only, especially if the topic describes reformulation in content? Moreover if reader cannot locate product in a certain group (vide remarks in ‘Food selection’) median is presenting poor data, with impossibility to correctly drawing a conclusions.

Differences between PAHO and CMR models will be more clear if authors would present both  models of food next to each other as well as beverages.

Discussion:

Line 263: Authors claimed that energy value of examined products increased during 2 years although in the body and figure1b there is decrease of sugar (calories carrier).  

Same with sodium (line 264) that in fig1b decreased from 55 to 52%.

Lines 265-268: Authors claimed that ongoing legislative debate over FOP warning labels did not decrease “unhealthly” substances in food. My concern is: if the law is not yet binding, it is difficult to expect that producers will apply it. Therefore, the examination of the available labels in the light of the probabilities of the following legal regulations seems to be methodologically ineffective.

Authors could also check if producers (manufacturing for and in Colombia) when export a product  abroad change its composition according to foreign market.

Line 272: What kind of comprehensive regulations authors mean?

Lines 329-332: Definitively it’s not correct. First authors examination point median only (as I gave an example of bun and package cake). Secondly food composition tables or refer to exact example of food (candy bar ”xx”) or narrowing the group (e.g. candy bars filled with caramel, milk covered in milk chocolate, with detailed composition and weight).    

line 342: how great this ”underestimation” was?

Reviewer 2 Report

Dear Authors,

I thank you for this very interesting paper, dealing with food labelling.

I have noticed:

line 117 (n=8): it is not clear what "n" means;

line 201-215: check the paragraph setting;

line 227: move table 2 in next page. Description of table must be in the same page with the table.

line 348: Conclusions have to be improved. 

line 409: references. Follow carefully the rules adopted by MDPI for this journal.

Reviewer 3 Report

Reformulation of Packaged Foods and Beverages in the Colombian Food Supply

Thank you for the opportunity to review this research. The authors provide an interesting analysis of food and beverage product reformulation in a subset of products available in large supermarket chains in Bogota, Columbia. The article is well-written for the most part, but I have outlined some suggestions below to improve and clarify some sections, particularly the methods.  

Introduction

Line 37 What do FOP labels warn consumers about?

Line 43 Suggest replacing “receiving” with another term to clarify that they are required to add FOP label, it isn’t something that is received and optional.

Line 50, Lines 67-68 Further discuss relevant policies adopted in Peru and Chile and how they might have had “spillover” effects in Columbia.

Materials and Methods

Lines 85-87 Describe the sample setting and context in greater detail. What is the population size of Bogota, Columbia? How do the sociodemographic characteristics of consumers in Bogota compare to the rest of the country?

Also describe the characteristics and representativeness of the five largest supermarket chains sampled from. Are they present in other cities?

Generally, is this a study sample that can produce findings that are generalizable to other areas in Columbia? Add strengths and limitations of this approach to the discussion as well.

2.1.1 Beverage Selection & 2.1.2 Food Selection

These sections are confusing as written. It may be helpful to show a flow diagram or present the information in a more streamlined manner to outline eligibility criteria and reasons for exclusion across stages and criteria.

Line 98 Why were other or private label brands excluded?  How many private label or unidentified brand products were excluded?  What implications does this have for the results?

Line 100 What is meant by “substitutes”?

Lines 127-129 What is meant by the total market share represented by each category? Is that the total share of sales of a food product category out of all products sold? Or is it supposed to convey the total market share of brands selected within a product category?  Please clarify.

Lines 130-131 Write out PAHO at first use and describe the PAHO and Chilean regulations in greater detail.

2.2 Nutrient Profile Systems

I suggest moving this section before the beverage and food selection sections because some of the selection criteria were based on the profile systems.

2.3 Statistical Analyses

Lines 188-190 Why were food and beverages compared separately and combined as an overall category? What is the reasoning behind this? It appears in the results that the significant differences for the overall category are driven by the beverage category, and do not add any particular value to the findings.

Line 199  Remove the p and less than symbol - alpha levels are set at a value and it is understood that anything less than that value is considered significant (e.g., alpha was set at 0.05).

Results

Remove overall food and beverage category results unless sufficient justification  can be provided for their inclusion.  

Lines 221-222 Remove this sentence or revise so that it does not say that median sugar content “rose” since the difference was not significant.

Lines 242-243 “ same as previous comment

Line 238 Replace “regulated” with a more specific term (e.g., labeled) as all products should be subject to some sort of regulation – this is referring to a specific action resulting from the regulation (FOP labelling).

Line 241 Revise to convey potential, not actual labelling (e.g., The mean number of FOP labels that would be required for beverages declined from…)

Discussion

Lines 261-272 Provide greater detail and relevance for the income status of Columbia and other countries mentioned in the context of this research. This may be relevant for the introduction.

Line 264 Is “food supply” referring to only foods in Columbia or all foods and beverages?

Line 321 Revise to specify that some beverages among major brands in a subset of Columbian retailers have undergone reformulation during the study period. This study uses data from one location in Columbia and may not reflect changes in other areas or for brands excluded from this analysis.

Author Response

Thank you for your thoughtful review. Please see the attachment.

Reviewer 4 Report

Reformulation is an important tool to improve population diets because it does not require an active change by consumers. This paper contributes to identifying the drivers (or lack of) for food and beverage manufacturers to improve the healthiness of products. Therefore I consider this a useful paper to publish.

There are some minor errors and I recommend the formatting of the results section is improved to ensure clarity.

Why were private label brands excluded? In some countries these could be an important contribution to sales. Perhaps sales data was not captured by Euromonitor.

Results: I think there are some errors with formatting of the first paragraph. I suggest having separate paragraphs for the results for each nutrient profiling system. Please check the correct table has been referred to. At the moment there are two Table B1s in the appendix. Figure 1 – rotate datapoints (99%, 50% etc) so easier to read. Line 238 – should 0.78 and 0.5 be 78%, 50%?

Minor errors:

83: regarded should be regard

207: Changes in saturated fat

Check in-text citations (spaces, before punctuation etc)

Appendix tables:

Left justify footnotes in appendix tables so easier to read

Less bold on table grid lines will make bolded values easier to spot

Reference McNemar’s test (second table B1)

Author Response

(The authors gave the same response as above.)

Round 2

Reviewer 1 Report

Manuscript have been improved, with answering to most of my concerns. It present interesting data that can be completely new point of view to Europeans (other legislation demands) or may be expanding knowledge of south America’s scientific.

Since Colombian’s low is not familiar to me I have concerns with using word HARMFUL (in lines 444 and 532) I have serious doubts if any government would allow to use harmful ingredients in food. I would suggest to change this word in doubtful, or unhealthy if over used ingredients.

Author Response

Dear Reviewer,

Thank you for your time and thoughtful review. Please see our response below.

Manuscript have been improved, with answering to most of my concerns. It present interesting data that can be completely new point of view to Europeans (other legislation demands) or may be expanding knowledge of south America’s scientific.

Since Colombian’s low is not familiar to me I have concerns with using word HARMFUL (in lines 444 and 532) I have serious doubts if any government would allow to use harmful ingredients in food. I would suggest to change this word in doubtful, or unhealthy if over used ingredients.

Our response: Thank you for your comment. We have changed "harmful" in line 532 (now 415) to "potentially unhealthy." In line 444 (now 336), we have added "potentially" before "harmful" and cite the example of trans fats. We believe it is important to keep the word "harmful," because trans fatty acids (TFAs) are widely known to cause harm to human health and have been banned or severely restricted in many countries. Currently, Colombia limits TFAs to 2% in oils/fats, but permits up to 5% of total fat from TFAs in foods.

"In order to generate substantive changes to the food supply, comprehensive regulations, such as mandatory FOP warning labels or bans on potentially harmful compounds like artificial trans fatty acids, may be required."

Reviewer 2 Report

Dear Authors,

You have improved very well your manuscript and now.

Kind regards

Author Response

Dear reviewer,

Thank you for your time and your thoughtful review. We look forward to addressing any further questions or concerns, should any arise.

Sincerely,
The authors

Reviewer 3 Report

Again, thank you for the opportunity to review this manuscript. The authors have fully addressed my comments in the review.

Author Response

Dear reviewer,

Thank you for your time and your thoughtful review. We look forward to addressing any further questions or concerns, should they arise.

Sincerely,
The authors